# Locomotion modulates specific functional cell types in the mouse visual thalamus

Çağatay Aydın [1,2], João Couto[1,2], Michele Giugliano [1,3,5,6,7], Karl Farrow[1,2,3] & Vincent Bonin [1,2,3,4]

The visual system is composed of diverse cell types that encode distinct aspects of the visual scene and may form separate processing channels. Here we present further evidence for that hypothesis whereby functional cell groups in the dorsal lateral geniculate nucleus (dLGN) are differentially modulated during behavior. Using simultaneous multi-electrode recordings in dLGN and primary visual cortex (V1) of behaving mice, we characterized the impact of locomotor activity on response amplitude, variability, correlation and spatiotemporal tuning. Locomotion strongly impacts the amplitudes of dLGN and V1 responses but the effects on variability and correlations are relatively minor. With regards to tunings, locomotion enhances dLGN responses to high temporal frequencies, preferentially affecting ON transient cells and neurons with nonlinear responses to high spatial frequencies. Channel specific modulations may serve to highlight particular visual inputs during active behaviors.

[1] Neuro-Electronics Research Flanders, Kapeldreef 75, 3001 Leuven, Belgium. [2] Department of Biology & Leuven Brain Institute, KU Leuven, 3000 Leuven, Belgium. [3] VIB, 3001 Leuven, Belgium. [4] imec, 3001 Leuven, Belgium. [5] Department of Biomedical Sciences, University of Antwerp, Antwerpen, Belgium. [6] Brain Mind Institute, EPFL, Lausanne, Switzerland. [7] Department of Computer Science, University of Sheffield, Sheffield, UK. These authors contributed equally: Çağatay Aydın, João Couto. Correspondence and requests for materials should be addressed to V.B. (email: vincent.bonin@nerf.be)

**B**rain state and behavioral context profoundly influence how animals perceive and respond to stimuli. Perhaps one of the most striking examples is that of "inattentional blindness" whereby observers fail to notice salient scene changes when attending to specific aspects. Indeed, at the neuronal level, activity in sensory areas co-varies with behavioral factors such as attention[1–5], arousal[6], reward[7], and movement[8]. These modulations may control the flow of sensory information in the brain[6], improve sensory representations[9–11], or reflect integration of signal from multiple modalities[12,13]. A critical question is how behavioral modulations impact the sensory processing performed by the neurons

Responses in the mouse visual cortex are strongly modulated by locomotor activity[8,14]. The effects on cellular responses are diverse[15–17] and correlated with genetic cell types[8,11,15,16,18]. However, the degree to which locomotion alters the response properties of sensory neurons is less understood. This is particularly important for vision, because locomotion is associated with visual motion flow, which changes markedly the statistics of visual inputs.

One possibility is that visual neurons adapt to these changes by modulating the neurons' visual tuning properties, thus highlighting specific features that occur during locomotion. In accordance, visual neurons can alter their peak temporal frequencies[14,19], size tuning[20,21], and show tuning for movement speed[21,22]. Another possibility is that locomotion changes the responsiveness of specific cell populations. Indeed, locomotion may specifically enhance V1 gains at high spatial frequencies[11] through local inhibition[18]. Nonetheless, if locomotion acts differentially on specific cell populations it would further support the hypothesis that functional cell types form parallel information channels in the visual system.

While the majority of visual inputs reach primary visual cortex (V1) through the dorsal lateral geniculate nucleus (dLGN), behavioral modulations are thought to be relayed through top-down circuits[23], local connectivity[24], and/or neuromodulatory mechanisms[25]. However, thalamic nuclei (in particular the dLGN and the pulvinar) have also been shown to carry locomotion and

contextual signals[13,21,26,27], suggesting that some of the modulations observed in the visual cortex might originate in the thalamus. Nonetheless, if thalamic modulations are non-specific, its impact on sensory coding could be negligible.

We investigated in head-fixed mice the impact of locomotion on the integration of spatiotemporal contrast by dLGN and V1 neurons. Measuring responses to stimuli of different spatial and temporal frequencies, we found that locomotion broadly increases dLGN and V1 responses to visual stimuli but has only a limited impact on response variability and correlations. We also found that locomotion increases of dLGN responses to rapidly varying stimuli and that it modulates the activity of cell populations with distinct receptive field and spatial tunings. These results indicate that behavior can influence visual processing through activity modulations of specific functional cell types These modulations may serve to highlight specific visual inputs to cortex during active behaviors.

## Results

**Locomotion modulates amplitudes of dLGN and V1 responses.** To investigate the impact of behavioral state on neuronal responses in the early visual system, we performed multichannel recordings in head-fixed running mice (Fig. 1). C57Bl/6 J mice ($n = 16$ mice) were implanted with a head fixation bar and trained to voluntary run on a treadmill (Fig. 1a). Visual responses of dLGN and V1 neurons with well-isolated spike waveforms were recorded with multichannel silicon probes (Fig. 1b; Supplementary Fig. 1a–b). Simultaneous recordings from dLGN and V1 neurons were obtained in about half of the experiments (16/28 sessions in 9/16 mice). The behavior consisted of alternations between high-speed movement (mean speed $13.6 \pm 11.9$ cm/s; median duration 6.4 s; $n = 23$ sessions) and pauses (speed < 0.25 cm/s, median duration 8.8 s) (Supplementary Fig. 1f, g, h). To assess behavioral and arousal states, we measured treadmill movement, eye movement, and pupil size using infrared eye tracking (Fig. 1c). Locomotion coincided with large pupil size fluctuations, rapid dilations upon movement

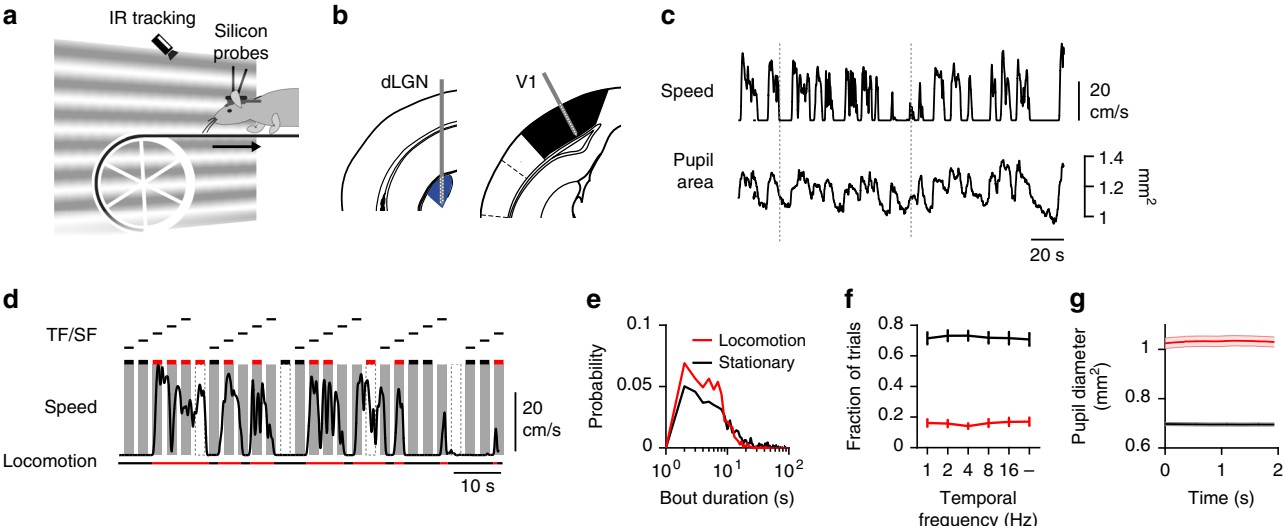

**Fig. 1** Experimental setup and behavioral paradigm. **a** Illustration of the linear treadmill assay. Full field, upward drifting sinusoidal gratings of different temporal (TF, 1,2,4,8,16 Hz) or spatial (SF, 0.01, 0.02, 0.04, 0.08, 0.16 cpd) frequencies were delivered to the right eye while animals ran on the treadmill. **b** Simultaneous multi-electrode recordings from dorsal lateral geniculate nucleus (dLGN, coordinates LM 2.1 AP 2.5) and primary visual cortex (V1, coordinates LM 2.5 AP 3.8). Coordinates in mm from bregma. **c** Locomotion speed (top) and pupil size (bottom) as a function of time. Scale bar, 20 cm/s. **d** Visual stimulation epochs (shaded areas) were categorized into locomotion (red) or stationary (black) trials based on locomotion speed measurements (black trace from c dashed lines). Scale bar, 20 cm/s. **e** Distribution of the duration of locomotion and stationary bouts (TF experiments: $N = 12$ mice in 23 sessions). **f** Fraction of locomotion (red) and stationary trials (black) for each temporal frequency (average±s.e.m. across sessions). **g** Pupil size as function of time for locomotion (red) and stationary (black) trials (TF experiments: 11/23 sessions with pupil size data; average ± s.e.m., $N = 335$ and 2414 epochs)

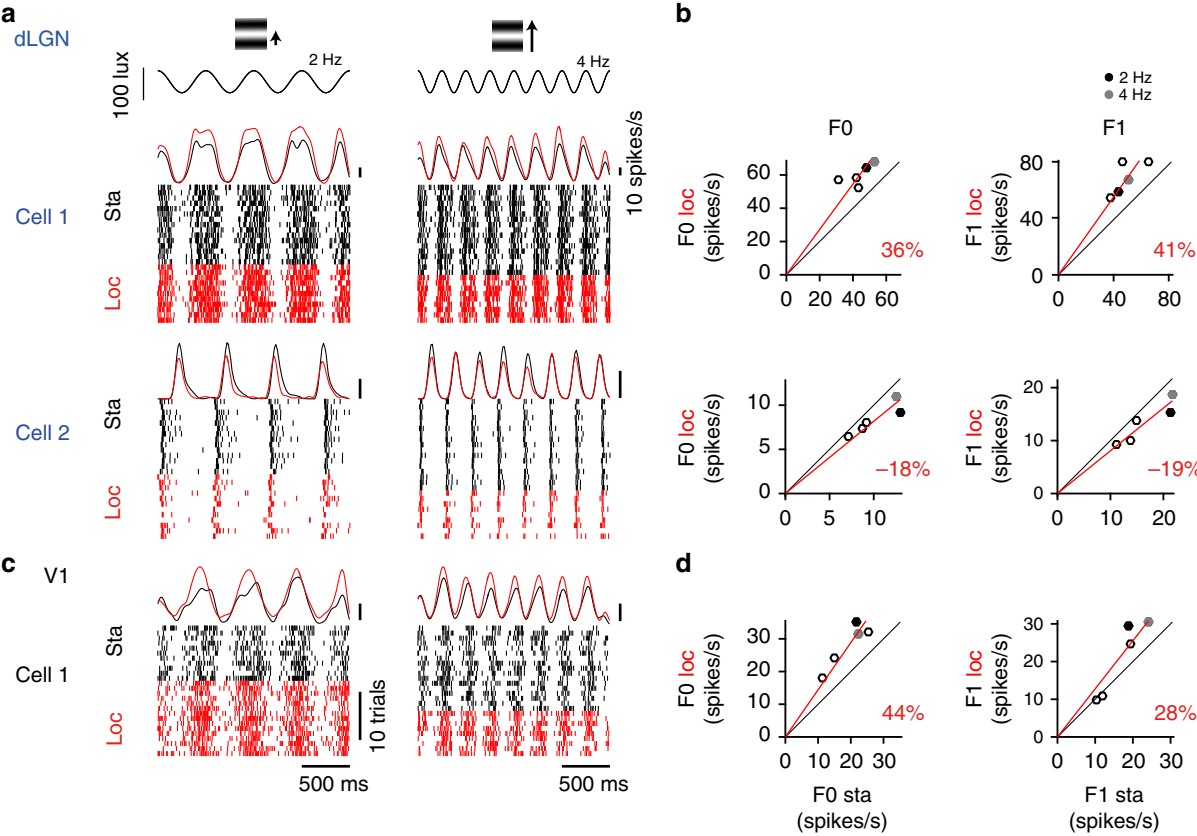

**Fig. 2** Locomotion related modulations of early visual responses. **a** Peri-stimulus time spike raster plots and histograms for two dLGN neurons in response to full field drifting gratings stimuli (temporal frequency: 2 Hz, left; 4 Hz, right). Responses in locomotion (red) and stationary (black) trials are plotted separately. Scale bars, 10 spikes/s and 10 trials. **b** Mean firing rate (F0) and response (F1) amplitude in stationary vs. locomotion epochs trials for the cells in (**a**) (36% increase and 18% decrease in firing rate; 41% increase and 19% decrease in response amplitude). **c** Spike raster plot and histogram for a V1 cell. **d** Same for a V1 cell (44% increase in firing rate; 28% increase in response amplitude)

onset and slow constrictions upon pause onsets (Fig. 1c, g), consistent with locomotor related fluctuations in arousal[17,21].

To investigate the impact of locomotion on visual processing, we examined responses to upward drifting gratings stimuli of different spatial and temporal frequencies (Fig. 1d). Stimuli were presented to the contralateral eye in 2-s intervals, independent of movement, and interleaved with epochs of equiluminant gray screen (see Methods; Visual Stimulation and eye tracking). We selected neurons with robust responses to the stimuli (mean peak-to-peak F1 amplitudes > 2.0 spikes/s, TF dLGN, 232/403 cells; V1, 110/255 cells; SF dLGN 107/167 cells; V1, 44/55 cells, see Methods). Responses were characterized by an overall increase in firing rate (F0; dLGN, 3.8 ± 8.8 spikes/s, $N = 119$ cells; V1, 5.3 ± 7.4 spikes/s; mean ± s.d.; $N = 47$ cells) and a periodic response at the temporal frequency of the stimulus (F1; dLGN, 8.4 ± 12.8; V1; F1, 4.2 ± 3.8 spikes/s; mean ± s.d.). We compared responses within locomotion bouts (>1 cm/s for >1.6 s of 2-s trial duration) to responses within stationary epochs (<0.25 cm/s for >1.6 s of 2-s trial duration) (Fig. 1e). We obtained 30–50 repeated trials for each stimulus and blank epoch, 17% and 70% of which fell within locomotion and stationary epochs, respectively ($n = 23$ recordings sessions in 16 mice, TF experiment, see Methods; Selection criterion for locomotion and stationary epochs) (Fig. 1f). Pupil size differed markedly across behavioral conditions (Fig. 1g). Unless stated otherwise, all results described below stem from these two data sets.

Locomotion strongly influenced the amplitudes of dLGN and V1 responses (Fig. 2). Changes in dLGN and V1 responses linked to locomotion showed as an overall scaling of firing rate

responses to the stimuli (Fig. 2; Supplementary Fig. 2, red vs. black). The effects of locomotion on dLGN and V1 responses were diverse, even amongst simultaneously recorded neurons. Some neurons showed an increase in response amplitude (Fig. 2a, c top; Supplementary Fig. 2a, c top). Other neurons showed no effect or a weak response reduction (Fig. 2a bottom; Supplementary Fig. 2a bottom). To quantify the modulations, we computed the average fractional change in amplitude of responses between locomotion and stationary epochs (Fig. 2b, d; Supplementary Fig. 2b, d, red lines), quantifying the effects on F0 and F1 responses (see Methods: Modulation index).

The strengths of modulations in dLGN and V1 were similar (Fig. 3). The distributions of modulation indices in dLGN and V1 were similar (Fig. 3a–d; Supplementary Fig. 3a–d). In both areas, response F0 and F1 modulations were highly correlated ($r = 0.74$ and 0.77, $n = 232$ and 110 cells, dLGN and V1, Supplementary Fig. 3i–j). No consistent change in F1 over F0 ratio was observed. These change in visually-evoked activity were paralleled by changes in spontaneous firing rates (Supplementary Fig. 4g–l).

Locomotion increased both neural firing responses (F0, dLGN, 20.0 ± 37.2%, and V1, 36.1 ± 45.9%) and the amplitudes of the oscillatory responses to the stimuli (F1, dLGN, 13.1 ± 34.7% and V1 26.7 ± 39.9%, mean ± s.d., $n = 232$ dLGN and 110 V1 cells). Amongst cells showing a response increase (MI > 0), the amplitudes of F1 responses in dLGN and V1 were 29.6 ± 27.7% and 37.5 ± 36.3% larger during locomotion, respectively (mean ± s.d., dLGN and V1, $n = 176$ and 56 cells) (Fig. 3g). Amongst cells showing a reduction in response (MI < 0), amplitudes of F1

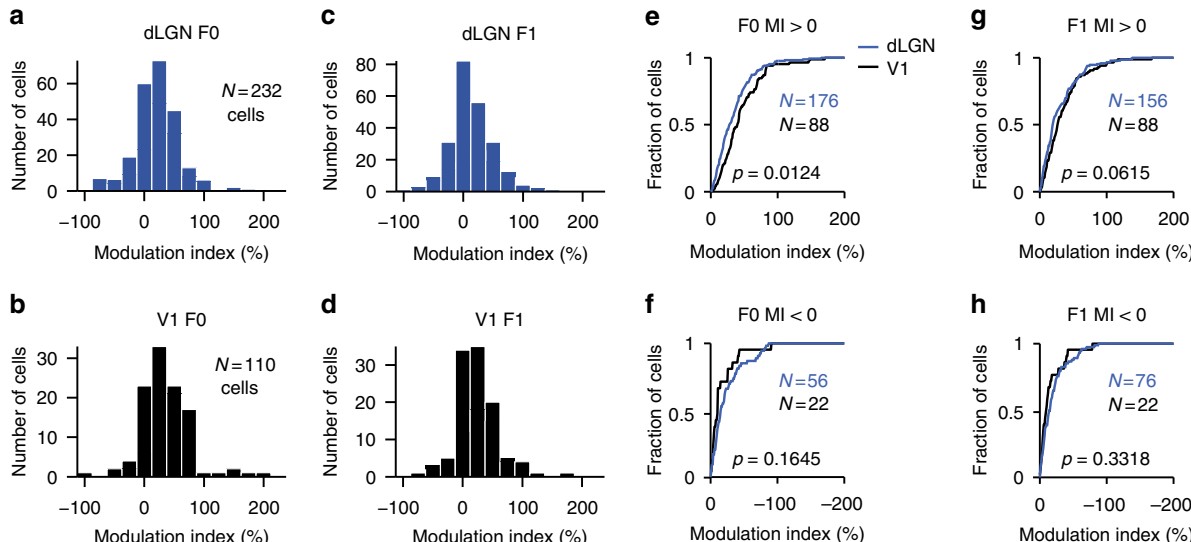

**Fig. 3** Similar response amplitude modulations in dLGN and V1. **a** Distribution of firing rate (F0) modulations for dLGN neurons (TF experiments: $n = 232$ cells). **b** Same as **a** for V1 neurons (TF experiments: $n = 110$ cells). **c** Distribution of response amplitude (F1) modulations for dLGN neurons. **d** Same as **c** for V1 neurons. **e** Cumulative distributions of firing rate (F0) modulations for cells with positive modulation index (MI > 0). **f** Same as panel **e** for negatively modulated cells (MI < 0). **g** Response (F1) modulations for cells with positive modulation index (MI > 0). **h** Same as **g** for negatively modulated cells (MI < 0)

responses decreased by $20.6 \pm 20.2\%$ and $16.5 \pm 18.9\%$ (Fig. 3h). While modulations of response F0 tended to be stronger in V1 than in dLGN ($p = 0.01$, MI > 0 and $p = 0.16$ MI < 0, K–S test, Fig. 3e, f), no pronounced difference between areas was observed for response F1 ($p = 0.06$, MI > 0 and $p = 0.33$, MI < 0, K–S test, Fig. 3g, h). Likewise, spontaneous firing rates in dLGN and V1 were significantly different between conditions (dLGN, $p = 0.01$ $t$-test; stationary: $13.6 \pm 1.1$; locomotion: $18.4 \pm 1.5$ spikes per second) (V1, $p < 0.001$; stationary: $17.5 \pm 1.3$; locomotion: $23.3 \pm 1.5$ spikes per second) (Supplementary Fig. 4g–l). Thus, locomotion modulates dLGN activity with modulation strengths similar to those observed in V1.

**Limited impact on response variability and correlations**. While dLGN and V1 visual response amplitudes increase during locomotion, concomitant changes in response variability could enhance or limit the gains. To investigate the effects of locomotion on trial-to-trial variability of responses, we selected recordings with at least ten repeated trials in either state and for each stimulus condition. To estimate firing rate variability of responses to the stimuli at short-time scales, we computed the Fano factor of responses in 50-ms bins and averaged the results over the stimulation epoch. To estimate response variability at longer time scales, we calculated the variance of F1 responses across trials and their coefficient of variation over the stimulation epoch.

Locomotion increased the strength of responses without increasing response variability of dLGN neurons (Fig. 4). While at short time scales, V1 firing rates showed a mild reduction in Fano factor (Fig. 4f, Supplementary Fig. 4f). No such reduction was seen in dLGN (Fig. 4e, Supplementary Fig. 4e). Thus, as firing rates increase during locomotion, there were no significant changes in variability at short time scales (Fano factor; dLGN, $p = 0.27$, V1, $p = 0.24$; K–S test; Supplementary Fig. 4e–f).

At longer time scales, the variance of F1 response amplitudes varied slightly in dLGN and significantly in V1 (F1 variance, dLGN, $p < 0.98$, V1, $p < 0.004$; K–S test) (Fig. 4c, d, Supplementary Fig. 4a–b). The nearly constant variance occurred in contrast with the pronounced increase in F1 response amplitudes (Fig. 4a, b). The increases in mean together with constant variance result

in a net reduction of the coefficients of variation of F1 responses (Supplementary Fig. 4c–d). Thus, as responses increase during locomotion, they do not become more variable.

To examine the degree to which variability is shared between neurons and how shared variability depends on behavioral state, we examined the correlation of spike counts from responses to the stimuli (Fig. 5a). We computed spike-count correlations between simultaneously recorded cell pairs in 1-s time windows starting 500 ms after stimulus onset and compared the results across behavioral states. While extracellular recordings can, in principle, resolve fast time-scale correlations such as those due to monosynaptic connections (Fig. 5b), short-time-scale correlations between cell pairs were rarely observed, with spike-count correlations extending over several hundred milliseconds (Fig. 5c, d). In dLGN, the distributions of spike count correlations during locomotion and stationary epochs were indistinguishable (Fig. 5e, g; $p = 0.12$; K–S test). In V1, there was a weak tendency for weaker correlations during locomotion (Fig. 5f; 0.08 in locomotion and 0.12 in stationary epochs; $p$-value 0.06; K–S test), consistent with previous reports[10,17,21]. Similar results were obtained using the full duration of the trials (Fig. 5g, h).

Thus, despite the strong enhancement of responses, dLGN and V1 neurons show no pronounced change in response variability and correlation across behavioral states.

**Impact on selectivity for spatial and temporal frequencies**. We next examined the impact of locomotion on response selectivity for spatial and temporal frequency (Fig. 6; Supplementary Fig. 5–6). In addition to modulating response gain, behavioral state may also affect neurons' receptive fields and how they respond to different stimuli. To address this question, we examined tunings of responses for spatial and temporal frequencies.

Consistent with previous work[28,29], dLGN and V1 neurons showed diverse tuning curves spanning a broad range of spatial and temporal frequencies (Fig. 6a, d, g, j). Locomotion broadly affected these responses (Fig. 6a, d, g, j, symbols, red vs. black). To quantify the impact on tuning, we fitted descriptive functions to the responses (Fig. 6a, d, g, j, curves) and extracted preferred spatial and temporal frequencies and tuning bandwidths

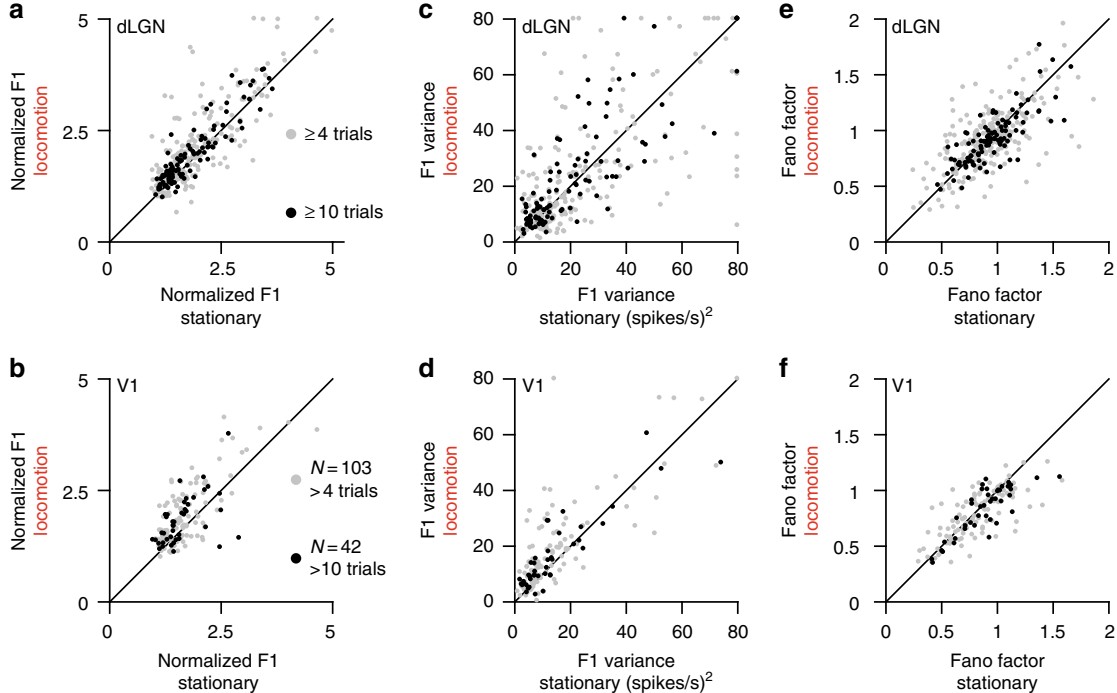

**Fig. 4** Weak impact of locomotion on response variability. **a** Normalized F1 responses of dLGN neurons in locomotion vs. stationary epochs for cells with ≥4 (gray) or ≥10 locomotion trials (black) (TF experiments: $n = 235$ cells and 106/235 cells, respectively). **b** Same as **a** for V1 neurons ($n = 103$ cells and 42/103 cells, respectively). **c** Trial-to-trial variance in F1 responses of dLGN neurons in locomotion vs. stationary trials (same cells as in **a**). **d** Same as **c** for V1 neurons (same cells as in **b**). **e** Fano-factor of firing rates responses in locomotion vs. stationary trials in dLGN (same cells as in **a**). **f** Same as **e** for V1 neurons (same cells in **b**). All measures are means computed across temporal frequency stimuli

(goodness of fit > 90%, $n = 143$ and 98 cells, dLGN and V1 in TF experiments, $n = 128$ and 47 cells in SF experiments).

The distributions of tuning parameters measured in locomotion and stationary epochs were very similar ($p = 0.31$ and 0.86 in TF experiments, $p = 0.25$ and 0.81 in SF experiments; dLGN and V1, $K$–$S$ test). The similarity held for preferred temporal frequencies (Fig. 6b, c, e, f; Supplementary Fig. 5a–d; Supplementary Fig. 6i–j), preferred spatial frequencies (Fig. 6h, i, k, l; Supplementary Fig. 5e-h; Supplementary Fig. 6k–l), and tuning bandwidths ($p = 0.77$ and $p = 0.16$ in TF experiments; $p = 0.81$ and $p = 0.94$ in SF experiments; dLGN and V1; $K$–$S$ test) (Supplementary Fig 5b–d, f–g).

To examine whether locomotion differentially affects responses to stimuli of different spatial and temporal frequencies, we computed the average ratio of responses in locomotion vs. stationary trials (Supplementary Fig. 6a–h). Locomotion affected responses to different spatial frequencies indiscriminately (Supplementary Fig. 6e–h, $p = 0.41$ for dLGN and $p = 0.67$ for V1, paired $t$-test of responses to 0.16 cpd in comparison to 0.02 cpd). However, dLGN responses to high temporal frequencies were enhanced during locomotion (Supplementary Fig. 6a–b, left panels, $p < 0.05$ at 8 Hz and $p < 0.001$ at 16 Hz, paired $t$-test in comparison to responses at 2 Hz), an effect that was restricted to neurons with positive modulation indices (Supplementary Fig. 6a–b, middle and right panels). This enhancement is similar to what was observed in visual cortex with calcium imaging[14]. Our sample of V1 neurons, however, did not show increased responses at high temporal frequencies but rather a tendency for weaker responses at 1 Hz (Supplementary Fig. 6c–d).

Thus, while locomotion has weak, unsystematic effects on the neurons' spatial tuning curves, it differentially affects population amplitudes of responses to different temporal frequencies.

**Modulations of dLGN functional cell types**. Rather than by changing the neurons' tuning properties, locomotion may impact visual coding by differentially modulating populations with distinct receptive field properties. To explore this possibility, we used $k$-means clustering to group dLGN neurons according to the shape of their temporal responses to the spatial frequency stimuli, using exclusively data recorded in stationary epochs. We then computed for each group the distribution of modulations indices from responses to spatial frequency stimuli in locomotion and stationary epochs. Cells showing suppression of activity by the stimuli instead of activation were excluded from this analysis ($n = 29$ cells).

The clustering yielded three broad groups of cells that differed in tuning for spatial frequency, response linearity and baseline activity (Fig. 7a, b, Supplementary Fig. 7c–f). One group with elevated firing responses at high spatial frequencies (Fig. 7a, b, Supplementary Fig. 7c–f, Group 1, $n = 35$ cells) showed particularly pronounced modulations (Fig. 7c, purple curve). By comparison, cells with responses tuned to mid-range spatial frequencies (Fig. 7a, Group 2, $n = 86$ cells) and cells with relatively high baseline firing rates (Fig. 7a, Groups 3, $n = 42$ cells) showed weaker modulations (Fig. 7d, e $p < 0.01$, significantly different from group 2 and 3, $K$–$S$ test). Notably, the elevation of firing at high spatial frequencies observed in Group 1 was not accompanied by periodic responses at the temporal frequency of the stimulus, indicative of nonlinear spatial summation as seen in Y cells in the cat retina and thalamus[30–32]. Other groups showed in comparison little indication of nonlinear responses to the stimuli.

The marked behavioral modulations observed of neurons in Group 1 are likely not a consequence of their elevated firing rate at high spatial frequencies. Pronounced modulations were observed in F1 responses over a broad range of spatial frequencies (Fig. 7c, top, Supplementary Fig. 7c, e). The differences in

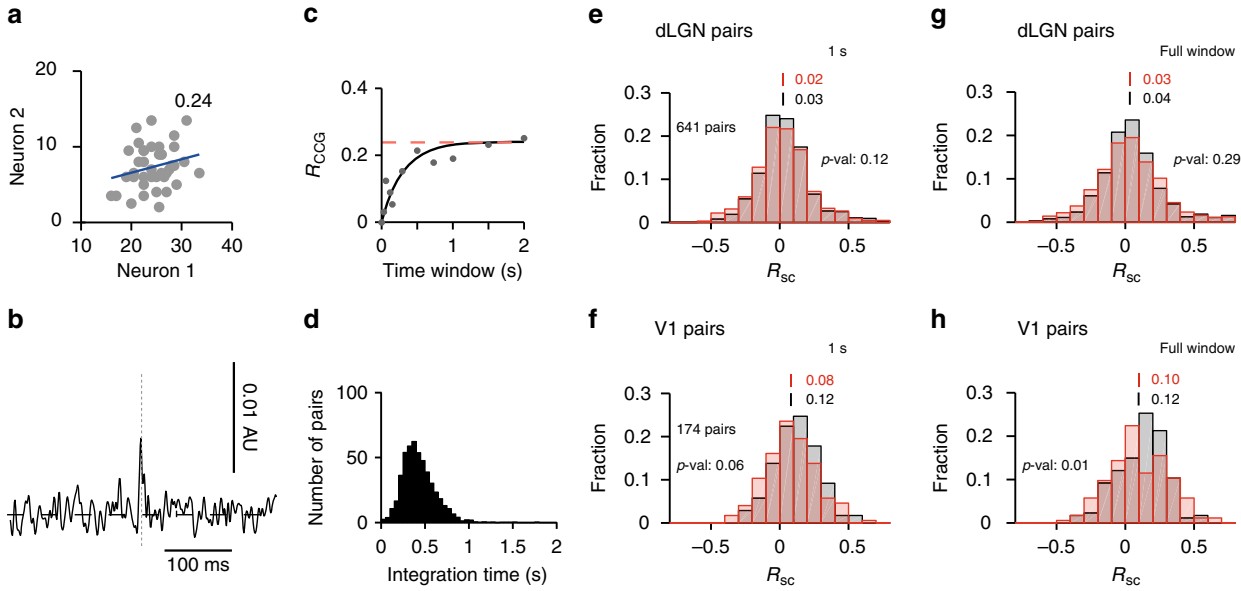

**Fig. 5** Weak impact of locomotion on pairwise activity correlations. **a** Trial-to-trial spike count responses of a correlated dLGN cell pair. Units are spike/s. **b** Example cross-correlogram (CCG) of a correlated dLGN cell pair. **c** $R_{CCG}$ (see Methods) for the pair in **a**. Red line indicate spike count correlation $R_{SC}$; black line is the exponential fit. Note how the $R_{CCG}$ converges to the spike count correlation ($R_{SC}$) for large window sizes. **d** Distribution of integration time from exponential fits to the $R_{CCG}$ for a sample of dLGN and V1 cell pairs. **e** Distributions of spike count correlations of dLGN cell pairs (641 pairs) in locomotion (red) and stationary (black) trials. Spike counts calculated using 1 s windows, starting 0.5 s after stimulus onset. **f** Same as **e** for V1 cell pairs (199 pairs). **g** Distribution of spike count correlations of dLGN cell pairs using the entire 2 s stimulus windows. **h** Same as **g** for V1 cell pairs

modulations were also not explained by the neurons' baseline firing rates (Supplementary Fig. 7d, f, top and centre). Finally, the distributions of inter-spike intervals at high firing rates (<10 ms) were comparable between groups suggesting that the response dynamic range was not saturated in any group (Supplementary Fig. 7b). The duration of locomotion and stationary bouts in the experiments did not explain differences between groups (Supplementary Fig. 7a).

To examine whether locomotion differentially modulates the responses of ON and OFF cells, we measured responses to uniform black and white stimuli alternating at 1 Hz and used responses in stationary epochs to categorize neurons into transient ON, sustained ON, transient OFF and sustained OFF (Fig. 8a, b, $n = 43$ cells, 33 cells, 24 cells, and 22 cells) (see Methods). We then compared modulation indices from responses to the spatial frequency stimuli in locomotion and stationary epochs (Fig. 8c, d). Neurons responding to both the ON and OFF phases were excluded ($n = 21$ cells). Transient ON cells showed more pronounced modulations by locomotion relative to other cell groups ($p < 0.01$, compared to other groups, K–S test, Fig. 8c, d, Supplementary Fig. 8). Thus, rather than indiscriminately impacting visual responses, locomotion preferentially modulates responses of dLGN neurons with specific visual response properties.

**Modulations of dLGN responses following atropine application.** To determine whether modulations of dLGN responses during locomotion reflect changes in light inputs due to changes in pupil size, we compared modulations of dLGN responses to spatial and temporal frequency stimuli before and after application of atropine to the contralateral eye (control vs. atropine, $n = 4$ mice) (Fig. 9). Clear modulations of responses by locomotion were observed after dilation of the pupil by atropine (Fig. 9a–c). Response modulations of dLGN neurons to temporal frequency stimuli were similar before and after atropine application ($n = 17$ cells and 42 cells; atropine vs control—same animals without

atropine application, $p = 0.24$; atropine vs baseline—neurons from Fig. 3b; 232 cells; $p = 0.55$, K–S test) (Fig. 9d, left inlet). Measurements from responses to spatial frequency stimuli, however, showed slightly weaker modulations ($n = 51$ cells and 41 cells; atropine vs control; $p = 0.61$; atropine vs baseline—neurons from Supplementary Fig. 3b; 164 cells; $p = 0.09$; K–S test) (Fig. 9d, right inlet).

To address whether differential modulations of ON and OFF cells reflect changes in light input, we compared the average responses of ON and OFF cells in locomotion and stationary trials before and after application of atropine (control; 4 mice, ON; $n = 27$ cells, ON-OFF; $n = 26$ cells, OFF, $n = 13$ cells, atropine; ON; $n = 25$ cells, ON-OFF; $n = 42$ cells; OFF, $n = 21$ cells). Similar locomotion-related modulations of ON and OFF cells were observed before and after atropine application (Fig. 9e). Therefore, while changes in light input due to pupil size fluctuations may contribute to locomotion-related modulations of dLGN responses, pupil size fluctuations do not appear to explain the differential impact on the responses of ON and OFF cells.

## Discussion

Using acute silicon probe recordings in head-fixed locomoting mice, we characterized the impact of locomotor activity on integration of spatiotemporal contrast by dLGN and V1 neurons. In both brain areas, neurons showed strong locomotor related modulations of response amplitudes and comparatively weak modulations of response variability and correlations. While locomotion has unspecific effects on dLGN and V1 neurons' spatial and temporal tuning curves, it enhances dLGN responses to high temporal frequencies. dLGN neurons with distinct spatial tunings also show differential modulations. These findings illustrate that behavioral modulations can affect sensory coding by modulating responses of specific functional cell types.

First described in the visual cortex[8], recent studies reported various locomotion-related modulations in dLGN[13,21,33,34]. While weak effects were also observed[34], effects on contrast[21] and

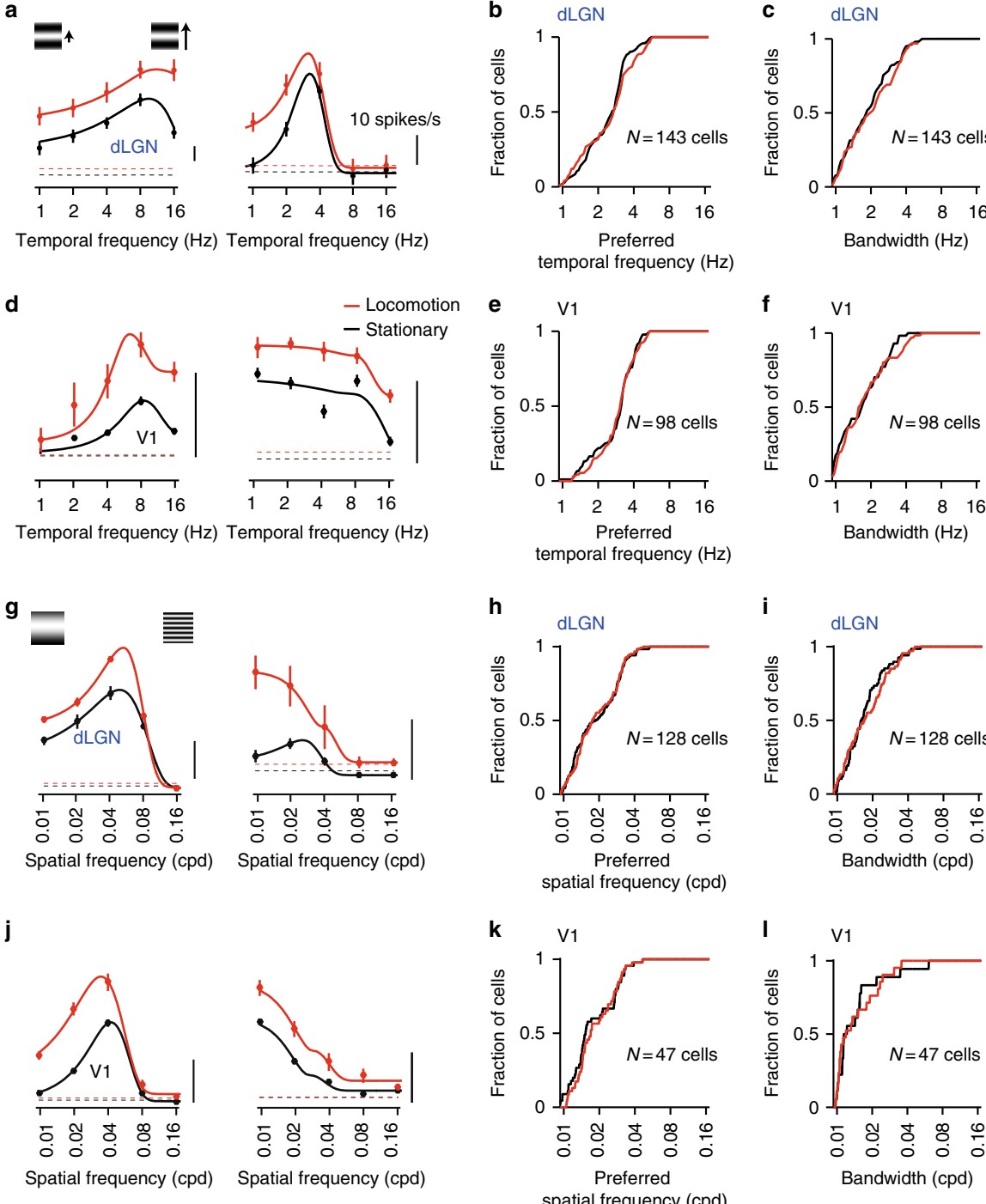

**Fig. 6** Unspecific impact on spatial and temporal frequency tunings. **a** Temporal frequency tuning of F1 responses of example dLGN cells in ocomotion (red) and stationary (black) epochs. Scale bar, 10 spikes/s. **b** Distribution of preferred temporal frequency of dLGN neurons ($N = 143$ cells) estimated from locomotion (red) and stationary (black) trials. **c** Distributions of temporal frequency bandwidth for the cells in **b**. **d** Temporal frequency tuning curves of F1 responses of example V1 cells estimated from locomotion (red) and stationary (black) trials. **e** Distributions of preferred temporal frequency of V1 neurons ($N = 98$ cells) in locomotion (red) and stationary (black) trials. **f** Distributions of temporal frequency bandwidth for cells in **e**. **g** Example spatial frequency tuning curves of dLGN cells. **h** Distributions of preferred spatial frequency of dLGN neurons (N = 128 cells). **i** Distributions of spatial frequency bandwidth for cells in **h**. **j** Example spatial frequency tuning curves of V1 cells. **k** Distributions of preferred spatial frequency of V1 cells (N = 47 cells). **l** Distributions of spatial frequency bandwidth for the cells in **k**. Dashed lines in d, g and j denote F1 responses to gray screen. Continuous lines in a denote F1 of shuffled spike times

sensorimotor integration[13] were reported. Our work differs from past studies in three ways. We assessed the impact on the neurons' spatiotemporal receptive fields. We examined modulations of populations with specific response properties. Finally, we used simultaneous recordings to directly compare activity modulations in dLGN and V1. Taken together, these measurements provide a detailed account of how locomotion influences the neurons' visual coding.

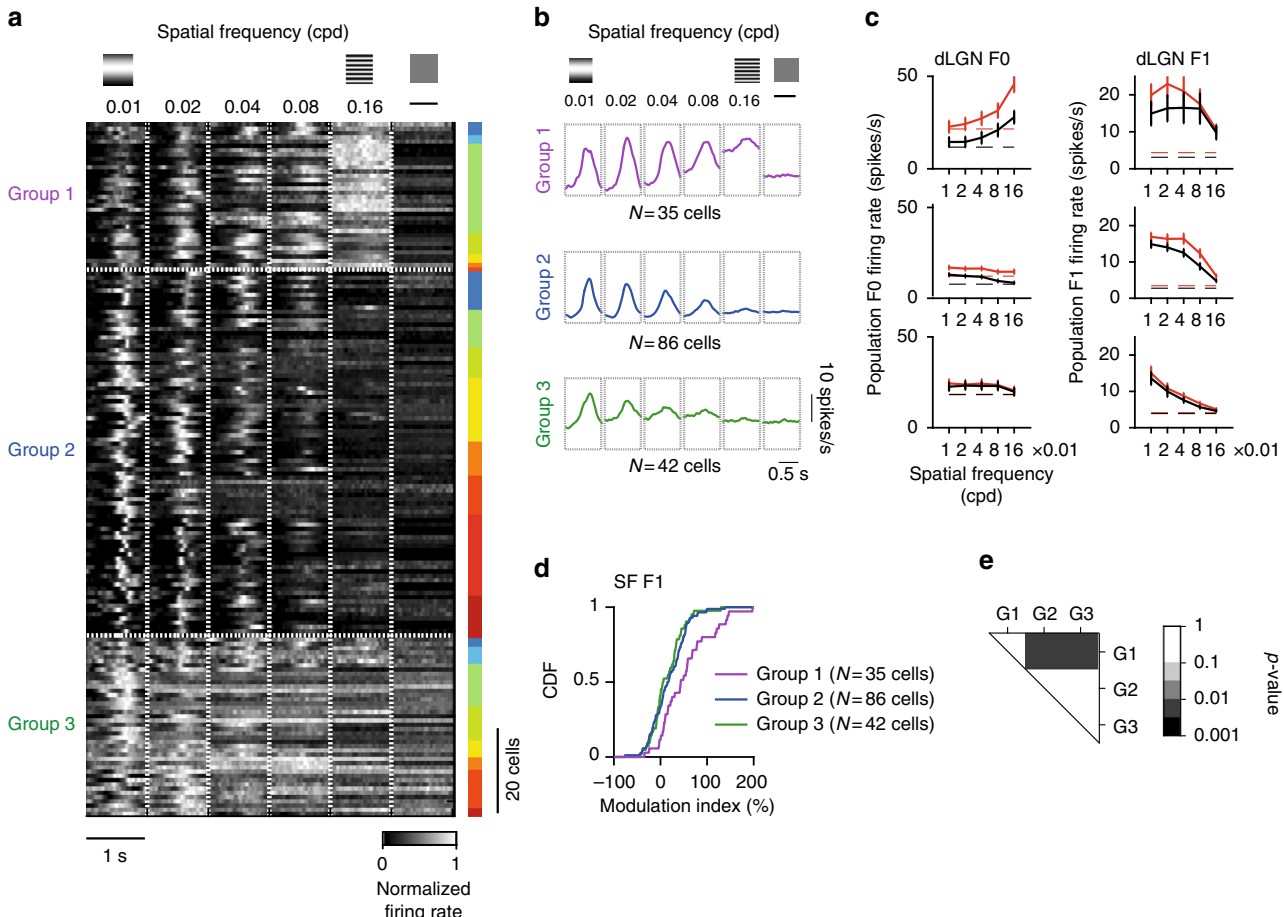

**Fig. 7** Preferential modulations of dLGN neurons with nonlinear responses to high spatial frequencies. **a** Grouping of dLGN cells ($N = 163$) based on normalized cycle averages to the spatial frequency stimulus and blank (30–50 trials). Cells were separated in three groups (Group 1, $N = 35$ cells, in purple; Group 2, $N = 86$ cells, in blue; Group 3, $N = 42$ cells, in green). Right inlet represents the correspondence of neurons to each experiment ($N = 15$ experiments, $N = 8$ mice). **b** Mean of the cycle averages for the groups in **a**. **c** Average F0 (left) and F1 (right) responses of cells in each group to the spatial frequency stimulus (same cells as in **b**). Scale bar for cycle averages is 10 spikes/s. **d** Response (F1) modulation index of individual groups to the spatial frequency stimulus. **e** P-values for the differences between groups computed using two-sample Kolmogorov–Smirnov tests

A calcium imaging study in V1 reported increased sensitivity to high spatial frequencies during locomotion[11]. The time course of the genetically encoded calcium indicator, however, precludes measurements of time varying responses to the visual stimulus, which are critical for the characterization of the neurons' visual properties. Using extracellular recordings, we could measure the linear and nonlinear components of the responses. Local inhibition through a specific class of interneurons has been proposed as a mechanism to modulate spatial frequency tuning in visual cortex during locomotion[18]. Nonetheless, our data suggests that increased firing rates in response to high spatial frequencies during locomotion in V1 might originate in the thalamus.

The properties of the subgroup we found to be most modulated resemble the responses of Y-type retinal ganglion cells observed in cats[30–32], and mice[35]. Like the neurons we identified, Y-type neurons may show transient response to visual stimuli[36]. Previous work noted the absence of transient ON cells in the mouse dLGN[29,37]. We observed cells with transient-ON, transient-OFF, sustained-ON and sustained-OFF response types (see also ref. [38]). This may reflect differences in sampling from the high density silicon probes we used. In our sample, most transient-ON cells had elevated firing rates at high spatial frequencies and showed higher modulations by locomotion than the other groups. This suggests modulations affect specific cell populations.

Locomotion was reported to reduce response variability and increase signal-to-noise ratio of responses in dLGN and V1[15–17,21]. We also observed increased response fidelity in dLGN and V1 during locomotion but in comparison to the pronounced impact on response amplitudes, the effects on response variability and correlation were minor. It has been proposed that the mechanism for the increase in signal-to-noise ratio is peri-somatic and dendritic inhibition[16]. Nonetheless, we found only a slight decrease in trial-to-trial variability during locomotion. While the impact of variability of thalamic inputs on V1 responses is not clear, it is possible that it affects how V1 neurons encode visual stimuli.

A previous study characterized pairwise correlations in spontaneous activity and found that these are reduced during locomotion in V1 but not in dLGN[21]. We found a similar behavior in correlations of responses to visual stimulation, whereby V1 neurons but not dLGN neurons showed a mild reduction during locomotion. We have computed noise correlations during the temporal frequency stimulus without taking the stimulus preference of the neurons into account. Further studies should investigate relation between visual response correlations and neuronal tuning during locomotor behavior[39].

Erisken et al.[21] reported that locomotion can impact size tuning of dLGN neurons, believed to reflect integration of spatial contrast. The preferences for spatial frequency and orientation of

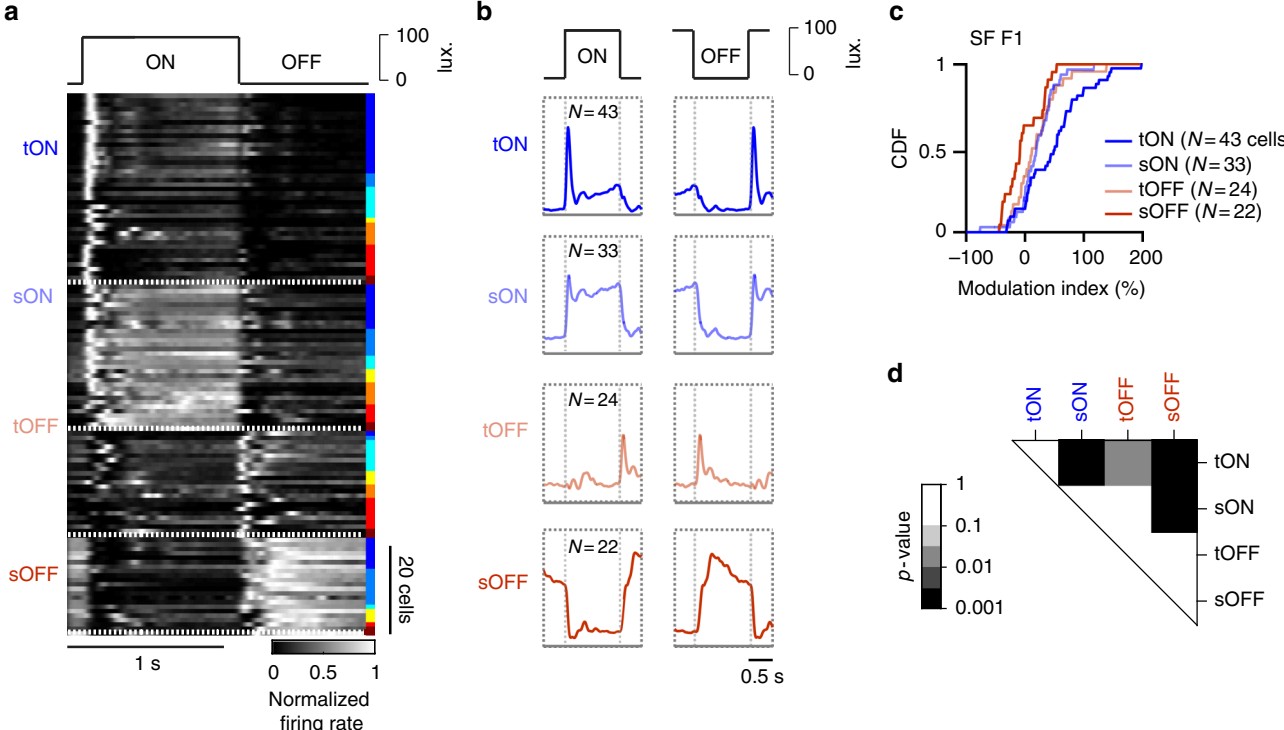

**Fig. 8** Preferential modulations of dLGN neurons with transient-ON responses. **a** Grouping of dLGN cells ($N = 122$) based on cycle averages in response to full-field contrast reversal stimulus (150–200 trials). Classification into four groups: transient ON, $N = 43$ cells (blue); sustained ON, $N = 33$ cells (light-blue); transient OFF, $N = 24$ cells (light red); and sustained OFF, $N = 22$ cells (red). Right inlet relates to which experiment each neuron was recorded ($N = 8$ experiments, $N = 5$ mice). **b** Mean of the normalized cycle averages over each group. **c** Cumulative distributions of response (F1) modulations of each group measured during spatial frequency sessions. **d** P-values for the difference between groups computed by K–S tests for F1 modulation

V1 neurons, however, seem largely preserved [11]. Accordingly, we found that locomotion neither affects the temporal nor spatial frequency tunings of the neurons.

In rabbits and rodents, arousal is associated with enhanced responsiveness[19] and locomotor activity[8]. While the effects of arousal and motor activity are intertwined, some effects seem to be specific to locomotion[17]. In our study we could not dissociate the contributions of arousal from those of locomotion, however, we provide evidence that the mechanism is independent of changes in light level that occur due to pupil fluctuations.

## Methods

**Animals, surgery, and histology.** All experimental procedures were approved by the ethical research committee of KU Leuven. Experiments were conducted in 16 male C57Bl/6j mice bred in the KU Leuven animal facility (22–30 g, 2–7 months old). Nine of these were used for simultaneous recordings from dLGN and V1. Dexamethasone (6 mg/kg I.M.) was injected four hours before the procedure. Mice were anesthetized with isoflurane (induced: 3%, 0.8 L/min $O_2$; sustained: 1–1.5%, 0.5 L/min $O_2$). The scalp was disinfected with %70 ethanol and Betadine and the skull exposed. The lateral and posterior muscles were retracted and Vetbond (WPI) was applied to exposed tissue and skin. Animals were then implanted with a custom-made titanium headpost, centered on the posterior left hemisphere[40]. A 2 mm diameter ground screw was then implanted through the skull in contact with the dura, over the cerebellum in the left hemisphere. Post-operative care was administered for 72 h following the surgery (Cefazoline [15 mg/kg I.M.] and Buprenorphine [0.6 mg/kg I.M.]). Mice were let to recover for one week and were habituated to the treadmill for 2–4 weeks. At least two days prior to the recordings, one/two ~1 mm craniotomies were made above the dorsal lateral geniculate nuclei (2.5 mm posterior to bregma, 2.1 mm lateral) or/and the V1 (3.8 mm posterior to bregma, 2.5 mm lateral). The dura was left intact. The craniotomy was covered with ACSF and a 5 mm circular coverslip. In 7/16 mice, artificial dura (Dura-Gel, Cambridge NeuroTech) was applied before covering the craniotomy. Finally, silicon sealant (Kwik-Cast, WPI) was used to cover the top of coverslip. Cefazoline (15 mg/kg I.M.) was administered to prevent infection in the 3 days following the procedure. Recordings were performed for up to 4 days following a 2-day recovery period from the craniotomy surgery. Between recording sessions, the craniotomies were covered in the same manner as described above. Probe tracks were

reconstructed from the last recording session by dipping the probes in Dil solution before insertion (Supplementary Fig. 1c). At the end of the last recording session, mice were anesthetized with ketamine (150 mg/kg I.M.) and perfused with phosphate buffered saline (PBS) followed by paraformaldehyde (4% PFA). The following day, brains were sectioned at 50 μm thickness using a cryostat (Leica, Germany). Slices were then stained with DAPI and imaged on a confocal microscope (Zeiss, LSM800).

**Head-fixed locomotion assay.** Headposted mice were placed on a linear treadmill apparatus[41]. The treadmill belt (150 cm long) was made of velvet paper or velcro tape (5 cm wide). Custom 3D printed wheels were located at both ends of the treadmill apparatus and a platform in the center. An optical encoder (200 or 500 pulse/revolution, Avago Technologies) was attached to one of the wheels and used to monitor animal velocity. A water reward was given at a fixed location (every 150 cm). A microcontroller (AT89LP52, Atmel) was used for driving the water reward valve (pinch valve—MS scientific). Encoder and reward pulses were logged with a data acquisition board (MCC) and stored for offline analysis.

Mice were water restricted 5 days after head-posting and habituated to head restraint for 2 days (10–30 min sessions). After habituation, mice were head-fixed on the linear treadmill apparatus for 30–60 min. Sessions were terminated in case of animal discomfort. Water rewards (~10 μl) were given every 150 cm. Animals were prepared for electrophysiology experiments when their performance surpassed 100 laps/h. Mice were trained with a gray screen (50% luminance), centered 20 cm away from the right eye. The average weight before water restriction was $26 \pm 3$ g. Mice were given 3 min of water access per day. If their weight dropped 15% of the weight before water restriction they were given free access to water.

**Visual stimulation and eye tracking.** Sinusoidal upwards drifting gratings (full-screen, 2 s duration) with varying temporal frequencies (1, 2, 4, 8, 16 Hz) and spatial frequencies (0.01, 0.02, 0.04, 0.08, 0.16 cpd) were displayed on a calibrated 22" LCD monitor (Samsung RZ2233). The screen was positioned in front of the contralateral eye covering 0° central to 120° peripheral and −15° lower to 25° upper visual field (Supplementary Fig. 1G). Data for temporal frequency was gathered from 14 animals and for spatial frequency from 13 animals. In 9/23 temporal frequency sessions and 9/15 spatial frequency sessions, stimuli were interleaved with 1 s epochs of equiluminant gray screen. In the remaining sessions, the gray screen was only presented at the end of the trial sequence. The movement and size of the contralateral pupil were monitored at 30 frames per second

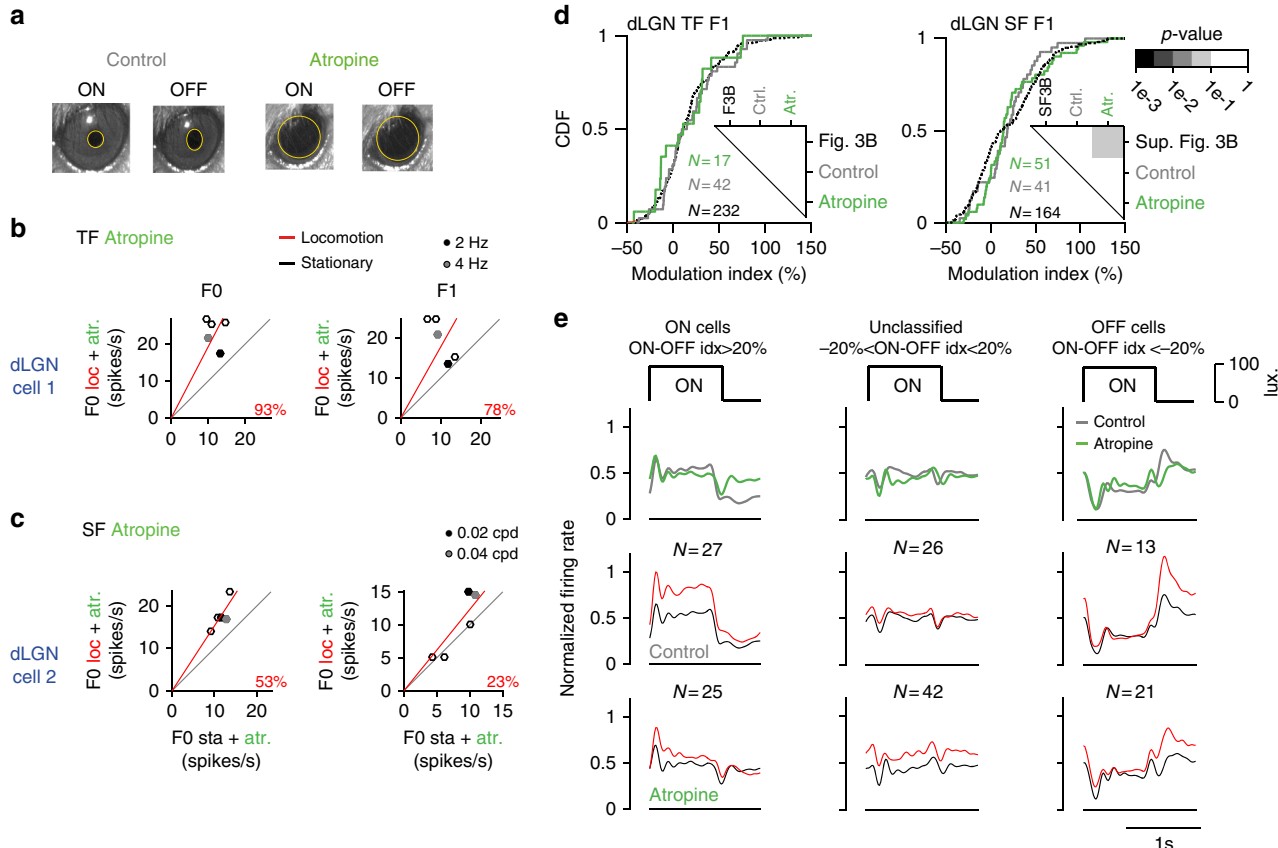

**Fig. 9** dLGN response modulations following atropine application. **a** Pupil diameter without (black, ON cycle/OFF cycle, left/right) and with (green) atropine recorded in the same animal (separate sessions). **b** Mean firing rate (F0) and response (F1) amplitude of an example cell in stationary vs. locomotion trials for five different temporal frequencies measured with atropine (93% increase in firing rate: 78% increase in response amplitude). High modulations by locomotion are independent of pupil diameter. **c** Same as **b** but for spatial frequencies (53% increase in firing rate; 23% increase in response amplitude). **d** Response (F1) modulation index of cells in Fig. 3g, h (black-dashed, $N = 232$ cells), control (gray $N = 42$ cells) and with atropine (green, $N = 51$ cells) to the temporal frequency stimulus (left). Response (F1) modulation index of cells in Supplementary Fig. 3g-h (black-dashed, $N = 164$ cells), baseline (gray, $N = 41$ cells) and under atropine (green, $N = 51$ cells) to the spatial frequency stimulus (right). P-values are shown in the inlet (K–S test). **e** Superimposed responses of neurons in the three groups ($N = 4$ mice; ON, unclassified and OFF) from the same animals with (Atropine) and without (Control) atropine application (top row) to full-field contrast reversal stimulus. Population mean of locomotion (red) and stationary (black) responses for the control group (middle row, baseline: ON cells: $N = 27$, unclassified cells: $N = 26$, OFF cells: $N = 42$). Population mean of locomotion (red) and stationary (black) responses for the atropine group (bottom row, baseline: ON cells: $N = 25$, unclassified cells: $N = 42$, OFF cells: $N = 21$)

(Supplementary Fig. 1F) with a CCD camera (AVT Prosilica GC660 with Navitar Zoom 6000) equipped with an infrared filter. Infrared light (700 nm) was directed at the eye to uniformly illuminate the pupil.

**Simultaneous dLGN and V1 recordings**. On the recording session, the Kwik-cast silicon elastomer was cleaned with 70% ethanol and removed. Craniotomies were rinsed and filled with ACSF. For simultaneous recordings, silicone probes (Neu-roNexus/Atlas Neuroengineering) were lowered using independent micro-manipulators (Scientifica). Probes were lowered under a stereoscope (Leica) to the target region at a speed of 10 μm steps per second and micro-positioned at ~1 μm per second. Single shank 16 channels poly-electrode probes (2 shaft, 16 channels, 50 μm site distance, and 375 μm recording area) were used for dLGN recordings and single shank linear 16 channels probes or 32 channels poly-electrode probes inserted perpendicularly to the surface of the exposed cortex for V1 recordings. The up-most electrode was used as reference and was located 100–200 μm below the pia in V1 (Supplementary Fig. 1d, e) and in white matter for dLGN recordings. No artifacts from spiking units detected by the reference electrode were observed. When the probes were in place, the craniotomies were covered with 1% agarose. Recording sessions were initiated 20–30 min after probe placement and lasted 1–1.5 h. We used a Grapevine (Ripple) or DigiLynx (Neurolynx) recording system to acquire electrophysiological signals at a sampling rate of 30 or 32 kHz respectively. The probes were cleaned with 1% enzymatic solution (Tergazyme) for 30 min at the end of each recording session. Recording sessions from individual brain regions, were performed using Neuropixels (Phase 2) probes with 120 recording sites[42]. Recordings were done using the ground screw as reference. For acquisition, we used the Whisper system (HHMI-Tim Harris) at a sampling rate of 25 kHz.

**Preprocessing and spike sorting**. Electrophysiological data was high-pass filtered (0.8–5 kHz) for spike detection and down-sampled to 5000 Hz for local field potential analysis by custom batch scripts based on Ndmanager plugins[43] {Jun, 2017 #87} (http://ndmanager.sourceforge.net). For A16 and A32 probes, channels were divided into four groups and spike sorting was performed on each separately. Spike waveforms were extracted and principle component analysis used to identify spike features and clustered using KlustaKwik (http://klustakwik.sourceforge.net). Manual refinement was done in Klusters (http://klusters.sourceforge.net). Only stable units that exibited a clear refractory period and had an isolation distance bigger than 20 were selected for further analysis. For Neuropixel recordings we used Spyking Circus for spike sorting[44] and Phy[45] {Yger, 2018 #96} for cluster refinement.

**Data analysis**. All data analysis was performed using custom-written code written in MATLAB (The Mathworks, Natick, MA).

**Selection of locomotion and stationary trials**. A trial was considered during locomotion when the animal was running at least 1 cm/s during 80% of its duration (2 s). Stationary trials were defined as those were the animal velocity was below 0.25 cm/s in at least 80% of the trial[21].

**Selection of visually-responsive cells**. The visual-evoked index was defined as the sum of Euclidian distances between mean response amplitudes (F1) and the mean permuted response amplitudes (F1-shuffled) of a neuron to each stimulus condition (TF 1, 2, 4, 8, 16 Hz; SF 0.01, 0.02, 0.04, 0.08, 0.16 cpd). Permuted F1 responses were computed by randomizing the order of inter-spike intervals of each trial/stimulus. The resulting spike train has the same number of spikes however does not maintain the temporal structure. Units were deemed visual if the average visual-evoked index across stimulus conditions exceeded 2.0 spikes/s, representing a robust visual response (F1) to least one stimulus.

**Modulation index**. To quantify the modulatory effect of locomotion, we computed the modulation index, defined as the difference between the unity line and the best fit of a single degree polynomial function, constrained at (0, 0), to the mean firing rate during quiescence and locomotion trials for each stimulus (Fig. 2b, d; Supplementary Fig. 2b, d for examples).

**Normalized F1**. In order to compare F1 responses from neurons with different firing rates, we computed the *normalized F1* defined as the F1 of the original responses divided by the F1 of the shuffled responses (see "Selection of visually-responsive cells" above) obtained by permuting the order of inter-spike intervals in each trial (Eq. (1)).

$$\text{Normalized } F1 = \frac{F_1}{F_1 \text{ permuted}} \quad (1)$$

**Tuning curves**. Responses to temporal frequency stimuli were fitted with a function composed of two-Gaussian described in Eqs. (2–3)[46].

$$R(\omega) = b_1 + (a - b_1) \times e^{-\left[\frac{p-\omega}{s}\right]^2} \text{ for } \omega < p \quad (2)$$

$$R(\omega) = b_2 + (a - b_2) \times e^{-\left[\frac{p-\omega}{s}\right]^2} \text{ for } \omega > p \quad (3)$$

where $R$ is the response amplitude ($F1$), $\omega$ is the temporal frequency, $p$ is the peak of temporal frequency, $a$ is the response amplitude at the optimal temporal frequency, $s$ is the Gaussian spread, $b_1$ is the baseline at low frequencies and $b_2$ is the baseline at high-frequencies.

**Spike-count correlations**. Noise correlations are a measure of trial-to-trial co-variability, however do not provide information on the time-scale of the variability. We computed the time-scale of the correlations during locomotion and visual stimulation using the $r_{CCG}$[47,48], defined in Eq. (4).

$$r_{CCG}(t) = \frac{\sum_{\tau=-t}^{t} CCG(\tau)}{\sqrt{\left(\sum_{\tau=-t}^{t} ACG_1(\tau)\right) \times \left(\sum_{\tau=-t}^{t} ACG_2(\tau)\right)}} \quad (4)$$

The CCG is the spike-train cross-correlogram (Fig. 5b) corrected with the shift-predictor (that is the CCG computed by shifting the neural responses of one of the neurons by one trial) to account for correlations induced by the stimulus.

Spike-train cross-correlograms ($r_{CCG}$) (Fig. 5c) revealed that a 1 s integration time window is sufficient to gather most correlations between neuron pairs (Fig. 5d, 450 ms). Spike-count correlations were computed by calculating the Pearson correlation between the responses of a neuron pair for each presentation of the same stimulus (see Fig. 5a, b). Only stimuli with a minimum of 10 trials were used. Jack-knife resampling (100 times) was used to estimate the correlation between pairs. Values were averaged across stimulus conditions.

**Cell grouping based on cycle responses**. To group cells based on their responses to the spatial frequency stimuli (Fig. 7), we computed cycle averages of responses for individual neurons from responses in stationary trials. Cells with reduced firing during visual stimulation were excluded ($n = 29$ cells). Cycle averages were convolved with a Gaussian kernel (2 ms half width) and normalized to the peak response. $K$-means was then used to group responses in three groups (Fig. 7, Groups 1, 2 and 3).

Grouping of dLGN neurons into transient ON, sustained ON, transient OFF and sustained OFF categories was done by applying K-means to cycle averages of responses to full field contrast reversal stimuli alternating at 1 Hz (Fig. 8). Neurons responding to both ON and OFF phases of the stimulus were discarded (21 cells).

**Analysis of pupil diameter and position**. Pupil diameter and position were extracted using custom-written code (mptracker, https://bitbucket.org/jpcouto/mptracker). The eye margins were manually identified and used to estimate the pixel-to-mm conversion assuming an average eye diameter of 6 mm. Parameters for morphological operations, filtering and threshold were adjusted manually for each dataset. Frames were smoothed with a Gaussian filter, and contrast adjusted using adaptive histogram equalization. In some cases, morphologic open or close

operations were used to mask artifacts caused by out-of-focus whiskers crossing the pupil. Frames were then threshold, contours extracted and fitted to an ellipse. Contours that did not resemble an ellipse were discarded. When multiple contours were present we selected the highest score based on the distance of the center of mass to the center of the eye and fit quality. The diameter was computed as the square root of the product of ellipse axis. Position was corrected by the corneal reflection (when present) and converted to spherical coordinates.

**Code availability**. All custom-written data analysis code are available from the corresponding author upon request.

## Data availability

All data reported in this study are available from the corresponding author upon request.

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

## Acknowledgements

We thank Dun Mao for experimental setup design, Adrienne Caiado and Frederique Ohms for animal training, Molly J. Kirk and Dun Mao for support with animal surgeries, and Karolina Socha for support with histology. MG acknowledges support from the Interuniversity Attraction Poles Program (IUAP) of the Belgian Science Policy Office, and the University of Antwerp. VB acknowledges support from FWO (Grant G0D0516N), KU Leuven Research Council (Grant C14/16/048). VB and MG acknowledge NERF Institutional Funding. NERF is funded by imec, VIB and KU Leuven.

## Author contributions

C.A., J.C., K.F., M.G., and V.B. designed the research; C.A. and J.C. built the experimental setups; C.A. acquired the data; C.A. and J.C. performed the data analysis with input from V.B. and K.F.; C.A., J.C., and V.B. wrote the paper with input from K.F. and M.G.

## Additional information

**Competing interests:** The authors declare no competing interests.

