## [Peer Review File · Nature Communications]

Reviewers' comments:

Reviewer #1 (Remarks to the Author):

This is a solid study reporting several important findings on the modulation of cells in dLGN and V1. The role of the thalamus in state modulation (or any cortical function) cannot be overstated and this study shows that already at the level of the dLGN there are clear modulations in firing that parallel the modulations in V1, which is a finding relevant to a large community of researchers working on these kind of mice models. There's some room for improvement and clarification nonetheless, but I recommend acceptance pending these minor revisions.

Major comments:

- * the results from Fig 5 deserve further exploration, by computing noise correlations on the firing rates of the entire trial as they are computed in e.g. Ecker et al. (2010). If not relevant please explain.
 - * the authors may have diluted their results by the way they define "stationary" and "locomotion" trials, in combination with the short running bouts, and possible anticipatory effects of locomotion on firing rates. If the data is available, it could be worthwhile to use more stringent criteria, or otherwise discuss this limitation when interpreting null findings, for instance when it comes to the lack of influence on correlations, as the arousal levels may not really differ between the conditions. If not the case they should argue why not.
 - * the authors could also discuss or address the same issue w.r.t. the modulations in dLGN.
 - * I might be misreading the paper but it seems the authors don't examine the change in spontaneous firing rate like e.g. Niell & Stryker (2010) or later studies on the same topic. How can we talk about signal-to-noise ratio or compare findings to those studies then? Maybe I'm wrong here but please explain.
 - * the effect of attention on firing in lower visual areas is not so clear, e.g. see Luck et al. (1997), but I don't know the latest word on this.
 - * The researchers don't describe the analgetic regime used, please describe. Antibiotics?
 - * Please describe the water restriction more in detail, what was the body weight the mice were kept at, how much water did they receive, etc.
 - * Dura must have been removed for insertion of probe?
 - * Algorithms for pupil detection?
 - * What was your reference electrode for dLGN and V1?
 - * Was the ground screw on the skull, through the skull, or in the skull?
 - * Recordings performed two days after craniotomies? How was the brain/dura protected in this period? Did you apply any chemicals?
 - * Great that you coat the probes, but would be nice to see at least one slice confirming the positioning of the probe in dLGN.
 - * Role of locomotion in behavior seems a little controversial, following some presentations at last SFN (Neske et al., talk from Jess Cardin) or work from McGinley et al. (2015) on auditory cortex. Don't want to be too much of a purist but "improving perceptual abilities" is not something one can measure, one measures behavior that is also governed by things like false alarm rates etc.
 - * "The dorsal lateral geniculate nucleus (dLGN) is the gateway of conscious visual information to the brain"
- Who knows...and mice.
- * Did your probe go straight down in V1 or at an angle
 - * What was the distribution of cells across layers
 - * Day/Night cycle?
 - * Why are you training the mice for so long and with water deprivation?
- I've seen mice run out of free will, after a 1-2 days habituation on the wheel.

* Where did your mice come from.

Minor comments

* "In N=9/23 sessions, the, stimuli was interleaved with epochs of 394 equiluminant grey screen"

Reviewer #2 (Remarks to the Author):

In this manuscript the authors analyzed the visual responses of dLGN neurons and their modulation by locomotion in head-fixed animals. They tested several previously proposed roles for this movement induced modulations. The authors confirmed that locomotion induced changes do not arise first at the cortical level but are present in the dLGN, contrary to early reports. They also report an absence of changes in pairwise correlations in V1 and dLGN with locomotion which had been previously described. The authors report that dLGN neurons that preferred low temporal frequencies are less modulated by locomotion. The authors also show that at the population level, locomotion does not shift the temporal spatial tuning responses of V1 nor dLGN, undermining the claim that visual responses are adaptively tuned for self-initiated movement detection. The analysis has been carefully performed and the conclusions are supported by the large amounts of data that were produced at high temporal resolutions. This could be a very important paper that contradicts several other studies.

Minor concerns

1) The authors should make a better case for what this study is important or interesting. Previous claims should be spelled out in detail and the contradiction with the current data should be emphasized. For example, the hypothesis that the visual filters are tuned for better detection of movement should be spelled out more clearly. This manuscript does not support this previous claim. The authors should spell out the types of non-linearities introduced by calcium imaging that might have led other authors astray. The same should apply respect to the claim that locomotion reduces correlations which implies reduced shared noise which is not supported by the data from this manuscript.

2) On figure 6 the authors show that there is no change at the level of the population in terms of spatiotemporal tuning with locomotion. The authors should add a figure showing how individual neurons change their properties in response to locomotion and how these changes are not shared across cells.

Reviewer #3 (Remarks to the Author):

In this study, the authors carried out a detailed characterisation of the effects of locomotion in mouse dLGN. More specifically, they report that the strength of locomotion effects varies according to functional cell type in dLGN, with the strongest increases in gain occurring for neurons with a preference for high spatial frequencies and transient ON responses. The authors conclude that locomotion-based response modulations in dLGN were of similar magnitude than those in V1.

Overall, this is a well executed study with clear and straightforward results. My main concern is whether this study has sufficient novelty and impact to be published in this journal. While the question under investigation is certainly of interest, we know, e.g., from several previous studies that neurons in dLGN are modulated by locomotion (Erisken et al., 2014; Roth et al., 2015; Williamson et al., 2015; Storch et al., 2017).

In general, and for Fig. 4 in particular (l. 142), I would recommend that the authors describe more

explicitly that they mainly present novel data for dLGN neurons. This can easily be missed in the current text, where some sentences are phrased in a very general way (e.g. "Locomotion increased the strength of responses without significantly increasing ...").

The statistical assessment of differential modulations by locomotion as a function of temporal frequency should be improved. Same holds true for the specific modulations of ON and OFF cells.

Please justify the usage of 3 groups for performing the clustering of dLGN neurons according to their cycle average. Please show the analyses excluding an explanation of differential locomotion-based response modulations based on baseline firing rates (l. 234). Please also show that for all the cell groups the locomotion behavior is identical, to rule out that some of these effects are driven by differences in running vigor or duration, rather than functional cell types.

MINOR

- l. 34, missing word?
- l. 123, V1 missing
- Comparison of modulation strength in dLGN and V1: since the comparison of the F0 is significant, and that of the F1 is almost significant, I would phrase the conclusion more carefully that "locomotion modulates amplitudes of visual responses in dLGN with modulation strengths rivalling those observed in V1"
- l. 141: variance of response F1
- l. 142: increasing response the variability
- l. 146: "However, these changes were modes": please quantify and state statistics
- l. 166: increase of the responses signal-to-noise ratio
- l. 190: could be change -> could be changed?
- l. 730: Ratio of locomotion trials by the stationary trials

NCOMMS-18-07250

Locomotion modulates specific functional cell types in the mouse visual thalamus

Cagatay Aydin*, João Couto*, Michele Giugliano, Karl Farrow, Vincent Bonin
Correspondence: vincent.bonin@nerf.be

We thank for the reviewers for the constructive comments and suggestions. We conducted additional experiments and performed new analysis to address the reviewer comments.

Summary of changes:

- 1) Rewrote Abstract, Introduction and Discussion to clarify significance of the work and relation to previous studies.
- 2) Added 2 animals to the atropine/control datasets and updated Fig. 9.
- 3) Extended the Methods sections with a section on "Analysis of pupil diameter and position".
- 4) Added a panel Fig. 1g to show how pupil diameter changes between behaviour conditions.
- 5) Included probe-track reconstructions for an example dLGN recording and the depth profile of V1 recordings to Supplementary Fig. 1c-e.
- 6) Added detailed analysis of the locomotion behaviour for average velocity and fraction of running in the trial to Supplementary Fig. 1f-g.
- 7) Included cumulative distributions for all panels in Fig. 4 to Supplementary Fig. 4.
- 8) Added tuning preferences of individual neurons during locomotion and stationary conditions to Supplementary Fig. 6.
- 9) Added a comparison of spontaneous and evoked firing rate to Supplementary Fig. 4.
- 10) Computed noise correlation for 2s window and added a panel to Fig. 5g-h.
- 11) Performed further statistical analysis on the distributions shown in Fig. 5, 7, 8 and 9.
- 12) Included a comparison of behavioural conditions, spontaneous firing rates, distributions of inter-spike intervals, tuning preferences for the 3 groups represented in Fig. 7 (in Supplementary Fig. 7).

Reviewer #1 (Remarks to the Author):

This is a solid study reporting several important findings on the modulation of cells in dLGN and V1. The role of the thalamus in state modulation (or any cortical function) cannot be overstated and this study shows that already at the level of the dLGN there are clear modulations in firing that parallel the modulations in V1, which is a finding relevant to a large community of researchers working on these kind of mice models. There's some room for

improvement and clarification nonetheless, but I recommend acceptance pending these minor revisions.

Major comments:

* the results from Fig 5 deserve further exploration, by computing noise correlations on the firing rates of the entire trial as they are computed in e.g. ¹. If not relevant please explain.

We now present measurements for 1 and 2-second windows (Fig. 5g-h). We recalculated noise correlations for isolated units showing stable responses (see Methods) and report statistical tests to highlight comparisons between conditions in the text. Enlarging the window to include stimulus onset transient had no significant effect on correlations in dLGN. We observed a small but significant difference between conditions in V1 in accordance with Erisken et al. 2014. We updated the main to reflect these results.

* the authors may have diluted their results by the way they define "stationary" and "locomotion" trials, in combination with the short running bouts, and possible anticipatory effects of locomotion on firing rates. If the data is available, it could be worthwhile to use more stringent criteria, or otherwise discuss this limitation when interpreting null findings, for instance when it comes to the lack of influence on correlations, as the arousal levels may not really differ between the conditions. If not the case they should argue why not.

* the authors could also discuss or address the same issue w.r.t. the modulations in dLGN.

The criteria used are in fact quite stringent (see Methods: "Selection of locomotion and stationary trials"). We now show movement speed and fraction of time spent running in Supplementary Fig. 1f-h for the two behavioral conditions. We also show how pupil size, a proxy for arousal, differs markedly between the conditions (Fig. 1g).

The pronounced changes in movement speed and arousal lead to pronounced changes in baseline firing rates (Supplementary Fig. 4) and visual responsiveness (Fig. 3). The limited impact on response variability and correlations despite strong modulations in dLGN are therefore quite surprising.

* I might be misreading the paper but it seems the authors don't examine the change in spontaneous firing rate like e.g. ² or later studies on the same topic. How can we talk about signal-to-noise ratio or compare findings to those studies then?

Good point! We added a direct comparison of the impact on spontaneous and visually evoked activity to Supplementary Fig. 3. We now note in the text that the

increase in evoked activity is paralleled by an increase in spontaneous activity. Additionally we have removed the statement about signal-to-noise ratio.

Maybe I'm wrong here but please explain.

* the effect of attention on firing in lower visual areas is not so clear, e.g. see Luck et al. (1997), but I don't know the latest word on this.

We included additional references and modified the Introduction to account for this comment.

* The researchers don't describe the analgesic regime used, please describe.

Buprenorphine (0.6 mg/kg I.M.) was injected 72 hours after head-posting; no analgesic was used during recordings. We added this to the methods section ("Animals, Surgery and Histology").

* Please describe the water restriction more in detail, what was the body weight the mice were kept at, how much water did they receive, etc.

We added this to methods ("Training").

* Dura must have been removed for insertion of probe?

The dura was left intact. The probe was inserted through the dura (see methods: "Animal, Surgery and Histology").

* Algorithms for pupil detection?

We now include a description of the algorithm for pupil detection in the methods ("Analysis of pupil diameter and position"). We posted the algorithm online: <https://bitbucket.org/jpcouto/mptracker>

* What was your reference electrode for dLGN and V1?

The up-most electrode was used as reference for Neuronexus/Atlas probe recordings. Neuropixels were done using the ground screw as reference.

We include additional information in the methods section ("Simultaneous dLGN and V1 recordings").

* Was the ground screw on the skull, through the skull, or in the skull?

The screw was implanted through the skull in contact with the dura. We added this the methods ("Animals, Surgery and Histology")

* Recordings performed two days after craniotomies? How was the brain/dura protected in this period? Did you apply any chemicals?

The craniotomies were filled with ACSF and covered with a 5 mm coverslip and silicon sealant (Kwik-Cast). In 7/16 experiments the craniotomies were filled silicone gel (3-4680, Dow Corning). This was added to the methods ("Animals, Surgery and Histology"):

* Great that you coat the probes, but would be nice to see at least one slice confirming the positioning of the probe in dLGN.

We added histological confirmation of dLGN probe placement from one example experiment to Supplementary Fig. 1c.

* Role of locomotion in behavior seems a little controversial, following some presentations at last SFN (Neske et al., talk from Jess Cardin) or work from McGinley et al. (2015) on auditory cortex. Don't want to be too much of a purist but "improving perceptual abilities" is not something one can measure, one measures behavior that is also governed by things like false alarm rates etc.

We removed "improving perceptual abilities".

* "The dorsal lateral geniculate nucleus (dLGN) is the gateway of conscious visual information to the brain"
Who knows...and mice.

This comment was taken into account when re-writing the Abstract and Introduction.

* Did your probe go straight down in V1 or at an angle

The probe was inserted straight down. This was added to the methods ("Simultaneous dLGN and V1 recordings").

* What was the distribution of cells across layers

We added the depth profile of the neurons recorded in V1 (Supplementary Fig. 1e). Additionally, we performed current source density analysis triggered by the onset of full screen contrast reversal stimulus to identify layer in V1 for an example (recorded with a neuropixel probe; Supplementary Fig. 1d). Neurons were recorded mainly from the granular (~450 micrometers) and infragranular layers.

* Day/Night cycle?

A standard 12h day/night cycle was applied in the animal facility, recordings were performed during the day.

* Why are you training the mice for so long and with water deprivation?
I've seen mice run out of free will, after a 1-2 days habituation on the wheel.

On the linear treadmill, mice do not run as much as on the wheel. Water deprivation helped mice habituation to head fixation in our hands. The treadmill provides a stable surface where mice alternate between stationary and "voluntary" locomotion bouts. We therefore see this as a feature, not a bug.

* Where did your mice come from.
Mice were bred in the KU Leuven animal facility.
We added this to the methods ("Animals, Surgery and Histology").

Minor comments

* "In N=9/23 sessions, the, stimuli was interleaved with epochs of 394 equiluminant grey screen"

The sentence is corrected.

Reviewer #2 (Remarks to the Author):

In this manuscript the authors analyzed the visual responses of dLGN neurons and their modulation by locomotion in head-fixed animals. They tested several previously proposed roles for this movement induced modulations. The authors confirmed that locomotion induced changes do not arise first at the cortical level but are present in the dLGN, contrary to early reports. They also report an absence of changes in pairwise correlations in V1 and dLGN with locomotion which had been previously described. The authors report that dLGN neurons that preferred low temporal frequencies are less modulated by locomotion.

The authors also show that at the population level, locomotion does not shift the temporal spatial tuning responses of V1 nor dLGN, undermining the claim that visual responses are adaptively tuned for self-initiated movement detection.

The analysis has been carefully performed and the conclusions are supported by the large amounts of data that were produced at high temporal resolutions. This could be a very important paper that contradicts several other studies.

Minor concerns

1) The authors should make a better case for what this study is important or interesting. Previous claims should be spelled out in detail and the contradiction with the current data should be emphasized.

For example, the hypothesis that the visual filters are tuned for better detection of movement should be spelled out more clearly. This manuscript does not support this previous claim. The authors should spell out the types of non-linearities introduced by calcium imaging that might have led other authors astray.

The same should apply respect to the claim that locomotion reduces correlations which implies reduced shared noise which is not supported by the data from this manuscript.

We have rewritten Discussion to clarify the relation to previous studies.

2) On figure 6 the authors show that there is no change at the level of the population in terms of spatiotemporal tuning with locomotion. The authors should add a figure showing how individual neurons change their properties in response to locomotion and how these changes are not shared across cells.

We agree. We added this figure (Supplementary Fig. 5, a-d temporal frequency, e-h spatial frequency).

Reviewer #3 (Remarks to the Author):

In this study, the authors carried out a detailed characterization of the effects of locomotion in mouse dLGN. More specifically, they report that the strength of locomotion effects varies according to functional cell type in dLGN, with the strongest increases in gain occurring for neurons with a preference for high spatial frequencies and transient ON responses. The authors conclude that locomotion-based response modulations in dLGN were of similar magnitude than those in V1. Overall, this is a well-executed study with clear and straightforward results.

My main concern is whether this study has sufficient novelty and impact to be published in this journal. While the question under investigation is certainly of interest, we know, e.g., from several previous studies that neurons in dLGN are modulated by locomotion (Erisken et al., 2014; Roth et al., 2015; Williamson et al., 2015; Storchi et al., 2017).

We agree. We revised Abstract, Introduction and Discussion to clarify the importance of the work and the relation to other studies. This is the first study addressing the impact of behavior on receptive field properties in the thalamus. While our results indicate receptive fields only change mildly, we show pronounced correlations between response properties and behavioral modulations providing an explanation for earlier work, and provide the first evidence of specific modulation of visual cell types.

In general, and for Fig. 4 in particular (l. 142), I would recommend that the authors describe more explicitly that they mainly present novel data for dLGN neurons. This can easily be missed in the current text, where some sentences are phrased in a very general way (e.g. “Locomotion increased the strength of responses without significantly increasing ...”).

We agree and have revised the manuscript accordingly.

The statistical assessment of differential modulations by locomotion as a function of temporal frequency should be improved. Same holds true for the specific modulations of ON and OFF cells.

This is an important point. We have now added statistical tests per condition on spatial frequency tuning (Fig. 7e) and to ON and OFF cells (Fig. 8e-f) and to control and atropine (Fig. 9d).

Please justify the usage of 3 groups for performing the clustering of dLGN neurons according to their cycle average.

We noted four broad types of responses to the SF stimuli (Fig. 7a).

- ❖ Suppressed by contrast, which were excluded from analysis (see Methods).
- ❖ Group 1: Cells with pronounced nonlinear responses to high SFs.
- ❖ Group 2: Cells with F1 responses tuned to midrange spatial frequencies.
- ❖ Group 3: Cells with higher baseline firing rates.

Our sample was not large enough for finer classification.

We added figure panels for the preference of F0 and F1 tuning (Supplementary Fig. 7e, g) and comparison of baseline firing rates. (Supplementary Fig. 7f, h).

We include further information about each group in the Results section and added a sub-section “Cell grouping based on cycle averages” to the Methods section.

Please show the analyses excluding an explanation of differential locomotion-based response modulations based on baseline firing rates (l. 234).

This was a typo. We meant to write that Group 1 and 2 have similar baseline firing rates but different modulation indices. We now show the bivariate cumulative distributions of spontaneous firing rates compared with peak F0 frequency for the groups (Supplementary Fig. 7h).

We also show that the groups have, at high firing rates, similar distributions of inter-spike intervals, which excludes differential saturation in firing rates across groups (Supplementary Fig. 7d).

Please also show that for all the cell groups the locomotion behavior is identical, to rule out that some of these effects are driven by differences in running vigor or duration, rather than functional cell types.

Duration of locomotion and stationary bouts were identical between groups. We added a supplementary figure (Supplementary Fig. 7c) with these results.

MINOR

- l. 34, missing word?

Fixed.

- l. 123, V1 missing

Fixed.

- Comparison of modulation strength in dLGN and V1: since the comparison of the F0 is significant, and that of the F1 is almost significant, I would phrase the conclusion more carefully that “locomotion modulates amplitudes of visual responses in dLGN with modulation strengths rivalling those observed in V1”

We agree with the reviewer and have rephrased the sentence more carefully.

- l. 141: variance of response F1

Fixed.

- l. 142: increasing response the variability

Fixed.

- l. 146: "However, these changes were modes": please quantify and state statistics

We added additional figures and quantification.

- l. 166: increase of the responses signal-to-noise ratio

Fixed.

- l. 190: could be change -> could be changed?

Fixed.

- I. 730: Ratio of locomotion trials by the stationary trials

Fixed.

References:

1. Ecker AS, Berens P, Keliris Ga, Bethge M, Logothetis NK, Tolias AS. Decorrelated neuronal firing in cortical microcircuits. *Science (New York, NY)* **327**, 584-587 (2010).
2. Niell CM, Stryker MP. Modulation of visual responses by behavioral state in mouse visual cortex. *Neuron* **65**, 472-479 (2010).

REVIEWERS' COMMENTS:

Reviewer #1 (Remarks to the Author):

Congratulations to the authors for this very nice work

Reviewer #2 (Remarks to the Author):

All my concerns have been appropriately addressed.